# MiRNAs Expression Modulates Osteogenesis in Response to Exercise and Nutrition

**DOI:** 10.3390/genes14091667

**Published:** 2023-08-23

**Authors:** Patrizia Proia, Carlo Rossi, Anna Alioto, Alessandra Amato, Caterina Polizzotto, Andrea Pagliaro, Szymon Kuliś, Sara Baldassano

**Affiliations:** 1Sport and Exercise Sciences Research Unit, Department of Psychology, Educational Science and Human Movement, University of Palermo, Via Pascoli 6, 90144 Palermo, Italy; carlo.rossi@unipa.it (C.R.); annaalioto374@gmail.com (A.A.); polizzotto.caterina8@gmail.com (C.P.); a.pagliaro91@gmail.com (A.P.); 2Centro Medico di Fisioterapia “Villa Sarina”, 91011 Alcamo, Italy; 3Department of Biomedical and Biotechnological Sciences, Section of Anatomy, Histology and Movement Science, School of Medicine, University of Catania, Via S. Sofia n°97, 95123 Catania, Italy; alessandra.amato@unict.it; 4Faculty of Physical Education, Józef Piłsudski University of Physical Education, 00-968 Warsaw, Poland; szymon.kulis@awf.edu.pl; 5Department of Biological Chemical and Pharmaceutical Sciences and Technologies (STEBICEF), University of Palermo, 90128 Palermo, Italy; sara.baldassano@unipa.it

**Keywords:** nutrition, macro- and micronutrients, bone health, exercise, exogenous and endogenous miRNAs, epigenetics

## Abstract

In recent years, many articles have been published describing the impact of physical activity and diet on bone health. This review has aimed to figure out the possible epigenetic mechanisms that influence bone metabolism. Many studies highlighted the effects of macro and micronutrients combined with exercise on the regulation of gene expression through miRs. The present review will describe how physical activity and nutrition can prevent abnormal epigenetic regulation that otherwise could lead to bone-metabolism-related diseases, the most significant of which is osteoporosis. Nowadays, it is known that this effect can be carried out not only by endogenously produced miRs, but also through those intakes through the diet. Indeed, they have also been found in the transcriptome of animals and plants, and it is possible to hypothesise an interaction between miRNAs produced by different kingdoms and epigenetic influences on human gene expression. In particular, the key to the activation pathways triggered by diet and physical activity appears to be the activation of Runt-related transcription factor 2 (RUNX2), the expression of which is regulated by several miRs. Among the main miRs involved are exercise-induced miR21 and 21-5p, and food-induced miR 221-3p and 222-3p.

## 1. Introduction

Epigenetic mechanisms may influence gene activity at the transcriptional and post-transcriptional levels [1,2]. Among the most studied epigenetic modifications, there are changes in the expression profile of microRNAs (miRNAs or miRs) [3]. Furthermore, it is already known that diet along with exercise, can modulate the expression of endogenous miRNAs and prevent or delay the development of some metabolic diseases such as bone disorders, confirming the importance of epigenetics in bone regeneration [4,5,6,7,8,9,10,11]. In addition, some foods contain miRNAs that after ingestion can influence the various biological processes [12,13]. The bone tissue is metabolically active and it is constantly remodelling in response to different stimuli [14,15,16,17]. MiRNAs expression together with specific transcription factors control the differentiation of the mesenchymal cells from which the osteogenic line cells originate [9]. They can therefore regulate multiple processes crucial for homeostasis, such as cell proliferation, differentiation, survival, and apoptosis and critically influence the regulation of different cellular processes and post-transcriptional gene expression, allowing cells to adapt to the environment [1,15]. Recent findings show that both exogenous and endogenous miRNAs may be able to withstand several critical conditions of the extracellular environment and circulate in body fluids because they are protected from degradation by ribonucleases by “packing” into lipid nanovesicles called exosomes [12,18,19,20,21] or associated with argonaute protein 2 (Ago2), which is the key component of the RNA-induced silencing complex (RISC) [1,12,18,21]. In humans, Ago2 possesses the activity of miRNA-guided mRNA cleavage or translational inhibition; it has also been demonstrated that it interacts with the mRNA-binding protein GW182, a mRNA-binding protein weighing 182 kDa, which is a marker for cytoplasmic foci referred to as GW bodies (GWBs) proposed to be cytoplasmic sites for mRNA storage and degradation [22]. Alterations in the expression of specific miRNAs, which can be collected from peripheral blood and used as biomarkers to study the organisms responses, are associated with an increased risk of some medical conditions such as fractures and bone disorders [15]. Several miRNAs are involved in the osteogenic differentiation of progenitor cells and influence some cellular processes in response to exercise and nutrition, as we will describe in the following sections.

## 2. Methodology Section

This article aims at understanding the possible epigenetic mechanisms influencing bone metabolism. To retrieve eligible articles, a manual search was conducted on the following databases: PubMed, Google Scholar and Web of Science. Articles published up to 31 March 2023 were included, with no limitation of the start of the research period. Research strategy adopted: (microRNA) AND (epigenetics) OR (nutrition) OR (bone health) OR (exercise) OR (osteogenesis). This research was extended using references within selected texts, which were considered. To summarize the findings of this review, we have divided the review into subsections. Two authors (PP and CR) conducted the literature search and ultimately resolved disagreements about including the article in discussions with a third researcher (SB). To identify duplicate studies, selected manuscripts from each database were loaded into EndNote software (EndNote version X8.1; Thompson Reuters, New York, NY, USA).

## 3. Exercise and Osteogenic miRNAs Expression

Bone tissue is a tissue that is sensitive to mechanical stimuli. Mechanical loads, including compression and deformation, are the stimuli that play essential roles in the differentiation and mineralization of osteoblasts, as well as maintaining high bone mass and density decreasing the risk of osteoporosis [23]. Some studies show that high-impact exercise increases bone mineral density (BMD) [9,24,25,26]. The load applied to the bone must always be gradual and administered considering the age of the subject and eventually, the pathologies that are in place. However, an in vitro experiment performed on osteoclast indicates that mechanical stimulus increases the expression of some miRNAs; these mechanisms could be considered important therapeutic candidates for the prevention and treatment of bone diseases, in particular for osteoporosis [27,28]. Recently, it has been shown that the half marathon increases the expression of miR-21-5p that promotes the proliferation of mesenchymal stem cells by targeting the two antagonists of the Transforming growth factor β (TGFb) pathway: Phosphatase and tensin homologue (PTEN) and Small mother against decapentaplegic 7 (Smad7) by reducing their expression. PTEN antagonizes the phosphatidylinositol 3-kinase (PI3K)/AKT/mammalian target of rapamycin (mTOR) pathway via lipid phosphatase activity and controls different cellular processes including survival, proliferation, energy metabolism, and cellular architecture [29]. TGFb is a pleiotropic cytokine that regulates many processes such as cell growth, differentiation, apoptosis, migration, and immune response. Smad7 is a TGF-β/Smad signal antagonist and is a negative regulator of Runt-related transcription factor 2 (RUNX2); it reduces PTEN expression, accelerates osteoblast differentiation and increases cell survival through PI3K/Akt signalling [9,30]. In fact, in vitro, PTEN-deficient osteoblasts proliferate faster by reducing apoptosis and increasing the cell size; this is consistent with the effects of activation of Akt and mTOR pathways [31]. Reduced expression of PTEN and SMAD7 induced an increase in the AKT/pAKT and SMAD4 protein levels, and this overregulates RUNX2 gene expression [9].

In addition, it has been seen that the proliferative potential of mesenchymal stem cells decreases with age due to the telomere shortening. During cellular division, it is possible to have a greater decrease in telomeres, with a consequent loss of genetic information, therefore an aging process that leads to cell death. Telomere length is therefore important to counteract the senescence of mesenchymal stem cells and telomerase plays an important role in this regard and in promoting the differentiation process [32]. Telomerase is an RNA-dependent DNA polymerase that synthesizes telomeric DNA sequences and provides the molecular basis for unlimited proliferative potential [33]. Oxidative stress in response to exercise increases the expression of the human telomerase reverse transcriptase gene (hTERT) in human marrow stromal cells (hMSC), increasing telomerase activity that promotes osteoblast differentiation [34].

Hua-Yu Zhu et al. have shown that the same miR-21 can regulate hTERT via PTEN in different processes [35]. Telomerase has also been reported to regulate autophagy with an important role in the differentiation and regulation of stem cells of various cell types, including mesenchymal stem cells [36]. The increased expression of telomerase may be due to oxidative stress resulting from physical activity. Autophagy would appear to be involved in the regulation of the cellular redox state induced by physical activity through the degradation of iron-binding proteins such as ferritin [36].

Furthermore, it is already known that resistance exercise induces an increase in iron levels, probably related to tissue damage. Excess iron is considered toxic precisely because of its ability to accept and donate electrons and to be involved in the reaction called “The Fenton reaction”, in which reacting with hydrogen peroxide generates both hydroxyl radicals and higher oxidation states of the iron evolving into free radicals [37,38].

The increase in osteogenesis-related genes expression following physical exercises such as RUNX2, muscle segment homeobox 1 (MSX1) and secreted phosphoprotein (SPP1) act to increase the expression of hTERT and telomere repeat binding factor 1 (TERF1), indicates the activity of mesenchymal stem cells and their increased capability to differentiate [34].

## 4. Endogenous miRNAs and Bone Metabolism

As we already said, the miRNAs are a group of small non-coding RNAs that regulate gene expression at the post-transcriptional level and cellular processes [39,40,41]. Notwithstanding most miRNAs are found inside cells, there is a large portion that are release in the body fluids such as blood, urine, saliva, seminal fluid, breast milk and other biological specimens in response to tissue damage, apoptosis and necrosis through active transport into exosomes or bonded to a protein as previously described [42]. Early studies have shown that about 10% of circulating miRNAs are secreted into exosomes, while the remaining 90% form complexes with proteins such as Ago2 [43,44].

These systems protect them from the degradation induced by ribonucleases and ensure their stability in body fluids; maybe this is the reason why they could play an essential role as biomarkers, because of their high specificity for the type of tissue or cell of the origin [1]. Recently, the interest in pointing out the miRNAs directly involved in osteogenic processes in response to exercise and nutrition has increased. A study performed by Vimalraj et al. utilizing mice has identified RUNX2 gene expression as the predominant factor required for osteoblast differentiation [14]. In particular, the isoform Type II of this gene has been found in undifferentiated mesenchymal cells, pre-osteoblasts and chondrocyte precursors e, in general, it is necessary for the differentiation of pluripotent mesenchymal cells to osteoblasts [45]. The osteoblasts differentiation is enhanced by the transcription factor Osterix (Osx), a downstream effector of RUNX2. Up to the present, several regulators such as signal transducer and activator of transcription 1 (Stat1), Twist Family BHLH Transcription Factor 1 (Twist) and Hes-related family bHLH transcription factor with YRPW motif 1 (Hey1) form a complex with RUNX2 resulting in osteoblast differentiation inhibitions. This is a fairly complex phenomenon as a critical balance between the differentiation of osteoblasts and the activity of osteoclasts to maintain bone homeostasis [14]. One of the most studied miRNAs is miR-21; the studies confirm the main role to promote stem cell osteogenesis through the Smad7-Smad1/5/8-RUNX2 and Akt pathways [9,15]. The over-expression of miR-21 acts as a promoter of osteogenesis by improving matrix mineralization and fracture healing [14]; Chen et al. highlighted the reduction in miR-21 levels in bone tissue and serum in osteoporosis patients [46]. The miR-21 inhibitions determine a reduction of the osteogenic capacity of human bone marrow mesenchymal stem cells (BMSCs) and calcium deposition by inducing osteoporosis [15]. However, the investigations on the miRNAs that can influence the RUNX2 expressions are constantly evolving [14]. In addition, as we will discuss below, the expression of osteogenic miRNAs is influenced by physical exercise, but also by nutrition.

## 5. Micronutrients Intake and Osteogenic miRNAs Expression

Nutrients can interact directly with the genome and indirectly through modulation of mechanisms, including DNA methylation, histone modification and non-coding RNA expression, in particular the miRNAs. Vitamins and minerals can induce the expression of miRNAs through the activation of transcription factors, which regulate the gene expression by induction of messenger RNA (mRNA) degradation or inhibit their translation. They can also modify the expression of DNA methyltransferases (DNMT) and different enzymes such as histone deacetylase and histone acetyltransferase that are involved in several processes such as transcription activation, gene silencing, DNA repair and cell cycle progression [47]. DNMT would appear to prevent demethylation in postmitotic neurons, which together with DNA methylation provide an epigenetic mechanism of gene regulation in neural development, function and disorders [48]. The modulation of the activity of these enzymes leads to changes in the methylation state of DNA as well as the histones, which in turn modulate the expression of some genes, including the miRNAs themselves.

### 5.1. Vitamin D Intakes

Among the micronutrients, vitamin D plays a key role in promoting bone mineralization [5,49,50], facilitating the intestinal absorption of phosphorus and calcium intake with the diet that is involved in bone calcification. Vitamin D is a member of the steroid hormones families with nuclear steroid receptors (NR) signalling function and is involved in the biogenesis and regulation of miRNAs expression [5,6]. It is already known that oestrogens have fundamental anticatabolic and anabolic effects on bones; therefore, the lack of oestrogen plays a central role in the development of osteoporosis [51,52]. The active form, 1,25 dihydroxy vitamin D (1,25(OH)2 D), can regulate the expression of osteoblastic mineralization factors by affecting the expression of some specific miRNAs like miR-637 and miR-1228. The miR-637 would seem to act by degrading the mRNA of Collagen Type IV α 1 Chain (COL4A1), whose expression, during osteoblastic differentiation, inhibits matrix mineralization; while miR-1228, is a mirtron, an alternative precursor for microRNA biogenesis that was recently described in invertebrates, that uses a different mechanism of action than classical miRNAs, as they bypass the cleavage of enzyme DROSHA, is exported out of the nucleus, split by endoribonuclease Dicer and incorporated into the RISC [53]. The miR-1228 reduces the expression of the Bone Morphogenetic Protein 2 (BMP2K) inducible kinase, a protein potentially implicated in cellular endocytosis and differentiation, but its molecular functions have remained unknown; it seems to be a potent inducer of bone formation through its stimulation of osteoblast differentiation [5]. Given the importance of vitamin D, it is therefore very important to guarantee high levels, which can be achieved through diet, such as cholecalciferol, thanks to animal foods intake such as salmon and blue fish or mackerel or cod liver oil consumed mainly as a supplement. Another source of vitamin D is vitamin D2 or ergocalciferol, which is a bit less active, but of which plant-based food is rich, for example, mushrooms. In any case, 80% of the vitamin D needed is guaranteed by sun exposure. Therefore, the best practice to increase vitamin D levels is to carry out physical exercise outdoors to guarantee good production and positive action also on bone metabolism [9,24].

### 5.2. Vitamin C Intakes

Even vitamin C plays a positive role in bone health; in fact, it is a cofactor in multiple biological processes such as collagen synthesis and antioxidant capability, regulating stem cell differentiation and improving osteoblast activity [6,8,54]. Clinical studies performed in humans and animals have shown that a deficiency of vitamin C leads to musculoskeletal alterations; noteworthy, the results showed that 100 mM of vitamin C effectively activates genes related to the musculoskeletal system in BMSCs, whilst lower doses, 25 mM, did not induce any effect. Similar results have also been observed for the regulation of vitamin C-dependent miRNAs production [8,55]. There is a possibility that vitamin C treatments regulate miR-29b-1 and miR-589-5p expression by promoting Octamer-binding transcription factor ¾ (Oct3/4), Nanog, sex-determining region Y-box 2 (SOX2) and Mitogen-Activated Protein Kinase Kinase Kinase 8 (MAP3K8) expressions in BMSCs contributing to cell proliferation and differentiation. In addition, it increases the expression of miR-371b-5p, miR-181a and miR-215 in BMSCs. The study hypothesis would be that these miRNAs, respectively, promote cell proliferation and differentiation of these cells [8,56,57].

### 5.3. Orthosilicic Acid (OSA) Intakes

Among the various micronutrients that contribute to bone health is also orthosilicic acid (OSA), which stimulates osteoblastic differentiation. Studies on ovariectomized rats orthosilicic acid fed compared to other deprived rats showed a higher bone mineral density (BMD) and trabecular thickness. Recently, it has been discovered that miR-130b plays a role in cell proliferation, differentiation, and apoptosis; its expression increased during the osteogenic differentiation of multipotent mesenchymal stem cells of the human bone marrow. The study provided by Yunhao You et al. found an increase in his level during osteogenesis in response to 20 mM of orthosilicic acid and suggested that overexpression of miR-130b promoted osteogenic differentiation. However, the mechanisms of action, which promote his transcription, have yet to be explored. Nutrition can contribute to the intake of OSA. Indeed, its absorption is more effective if taken from the liquid phase in which it is dissolved and easily assimilated not having to undergo major changes [11]. Regular mineral water contains about 6.8 mg per litre, but there is some water whose silicon content can range from 14.4 mg up to 60 mg per litre. In food it is mainly contained in the leathery parts of vegetables; for this reason, it is important to consume vegetables and fruits with peel and legumes and prefer foods that have undergone few industrial processes to preserve their structure and content.

### 5.4. Other Micronutrients Intakes

Recent studies suggest that other micronutrients, such as natural phenolic acids, usually found in plants that are commonly intake by diet, may have an important bone anti-resorption activity [4,58,59]. One of these compounds is syringic acid, 3,5-dimethoxy-4-hydroxybenzoic acid (SA), a phenolic acid that acts on mouse mesenchymal stem cells (mMSC) cells inducing the differentiation of osteoblasts. It increases miR-21 expression, which reduces Smad7 activity by targeting the TGF-b/BMP signalling pathway, resulting in increased RUNX2 expression, thus leading to the expression of osteoblast differentiation markers genes such as alkaline phosphatase (ALP), Collagen Type I α 1 (Col-I) and osteocalcin (OCN) in BMSCs [60]. Also, isoflavones, i.e., syringe, genistein, laminarin, hesperetin and sulphurin, promote osteoblast differentiation through activation of the BMP2/SMAD5/Akt/RUNX2 pathway [61]. Deep SA has several positive effects on bone health due to its strong antioxidant activity and also through antihypertensive, antiproliferative, antiendotoxic, antitumor, hepatoprotective and antihyperglycemic effects. It is mainly present in olives, walnuts and dates, but also in blue-coloured fruits and berries, where it is formed by the decomposition of lenin, an anthocyanin, and its aglycone, malvidin [62]. This justifies its presence in red wine (0.27 mg/100 mL) and red vinegar (0.30 mg/100 mL) whilst in dried fruits such as walnuts or peanuts as well as in olives, cocoa pulp, pumpkin, durum wheat and in smaller quantities in peas and cauliflower, the concentration can reach 33 mg/100 g [11].

## 6. Macronutrients Intake and Osteogenic miRNAs Expression

Macronutrients influence important signalling pathways that regulate human metabolism [63]. However, it is not effortless to discriminate whether a nutrition–gene interaction is the result of a direct or indirect effect due to the involvement of several bioactive components. In other words, nutrients can induce epigenetic changes, either through methylation of DNA or changes in some miRNAs expression. Carbohydrates or Carbs (CHO), proteins and fats are broken down during digestion into the composition monomers respectively monosaccharides, amino acids and fatty acids [10,64]. Few studies have been carried out on the role of carbs and lipids on bone health. Currently, Kang Gan et al. investigated the roles of miR-221-3p and miR-222-3p, in regulating the osteogenic differentiation of BMSCs under high blood glucose conditions. The results showed increased expression of these two miRNAs in the bone tissue of diabetic mice, inhibiting osteogenic differentiation via the IGF-1/ERK pathway activation [65]. Further studies showed that high blood glucose levels, together with increased free fatty acids (FFA), increased the expression of miR-449, which inhibits osteogenic differentiation of BMSCs through suppression of the Sirt1/Fra-1 (Fos-related antigen) pathway [66]. MiR-449 directly targets sirtuin 1 (Sirt1) of the SIRT family by binding the 3′-UTR sequence. Sirt1 belongs to the NAD+-dependent enzymes classes that catalyse the deacylation of acyl-lysine residues that regulate the life span of mammals, cellular energy metabolism and the balance between osteoclastic and osteoblastic activity through different signalling pathways [67]. A study reported that Sirt-1 activated by resveratrol (a flavonoid) inhibits osteoclastogenesis; also 1 Fra-1, a protein belonging to the activator protein 1 (AP-1) family of transcription factors, plays an essential role in osteogenesis. Increased expression of miR-449, also significantly reduced mRNA and protein expression levels of osteogenic-differentiation-related marker genes, including RUNX2, bone sialoprotein (BSP), collagen I, and OCN. The latter regulates bone mineralization and bone turnover in the late stages of osteoblast differentiation, whilst BSP is involved in the mineralization [66].

Some studies suggest that a diet containing low amounts of methionine (belonging to the sulfur amino acids, SAAs) increases the expression of the miRNAs that alter RUNX2 expression by altering bone structure in mice. This is important because the methionine cycle generates S-adenosyl methionine (SAM), a coenzyme used by DNA methyltransferases to methylate the histones and regulate the gene expression. Particularly, these studies demonstrated that mice fed low-methionine food, compared to control mice, had high expression of miR-204 in the bone marrow; this miRNA regulates Osterix and RUNX2 in bone, inhibiting the amount and function of osteoblasts and by inducing bone fragility. Maternal bone mass decreases during lactation since skeletal calcium is released into breast milk. Although renal calcium excretion is reduced with increased tubular reabsorption and this is not sufficient to prevent bone loss. During lactation there is evidence of increased parathormone-related peptide (PTHrP) homologs with the N-terminal fragment of parathyroid hormone (PTH) produced by the mammary glands, which plays a key role in increasing blood calcium concentration and, in combination with low estradiol levels, leads to high rates of bone resorption. However, weaning triggers skeletal recovery that occurs very rapidly after the end of lactation [68].

Regarding the effects of the proteins, a study carried out on maternal nutrition has shown that a low-protein diet negatively regulates mother and child bone mass; however, there are no studies investigating the epigenetic effect of a high-protein diet.

Ioannis Kanakis et al. showed that there was a correlation between bone mineralization and the level of protein dietary intake during lactation in mice; the expression of RUNX2, as well as Alp and Col1a1, were all decreased mostly in mice with a low-protein intake diet compared to the control mice. This indicates a direct correlation resulting in decreased osteoblastic differentiation and activity, particularly in miR-26a, 34a and 125b expression. In deep, miR-125b normally regulates the osteogenic differentiation of human MSCs whilst miR-26a reverses the bone regeneration deficit of MSCs and miR-34a inhibits osteoclastogenesis. The main pathways concerned appear to be the Wnt and IL-6 signalling pathways [69]. Therefore, protein malnutrition increases bone loss, and slows down and delays bone recovery.

Fully understanding the mechanisms could lead to draft nutritional guidelines for improving bone health. However, the effects of dietary proteins on bone health must be considered according to age, health, the diet habits of the population and exercise practice [70]. Certainly, it is well known that a high protein intake increases urinary excretion of Calcium (Ca) and on average is estimated that at least 1 mg of Ca is excreted for each additional gram of protein consumed [71]. This relationship is mainly attributable to the metabolism of sulphur amino acids contained especially in animal proteins and some plant proteins, resulting in increased acidity buffered by organic calcium release from the skeleton. The effects of proteins on bones can also depend on the intake of foods rich in calcium and alkalis, such as fruits and vegetables. A low protein intake reduces insulin-like growth factor production, which in turn hinders calcium and phosphate metabolism, bone formation, and the promotion of satellite cell activation [72]. These effects probably depend on the amount and type of protein and influence bone health through epigenetic mechanisms [73].

## 7. Exogenous miRNA and Bone Metabolism

So far, we have focused on the interaction between macronutrients and endogenous microRNA production and how this can induce epigenetic changes that affect bone health. New scientific evidence suggests that microRNA content in foods can survive the processes of digestion, absorption, and transport in the biological fluids by regulating the expression of some genes correlated with different biological fluids [18,74,75,76]. Food-derived microRNAs could act as a new functional component of foods, such as vitamins and other nutrients, that can influence human health [6,12]. It is important to consider the typical specificity of miRNAs depending on the kingdom of belonging. Often, miRNA genes remain the same between species and kingdoms throughout evolution; however, it is possible that their nucleotide sequences may change, which makes interspecies influence difficult. However, recent studies show that many human miRNAs share identical sequences with miRNAs from different species, demonstrating a possible influence on humans [18,77,78]. Nevertheless, a plant-based miR-168a has been detected in human serum and regulated the expression of the mammalian gene low-density lipoprotein receptor adaptor protein 1 (LDLRAP1) involved in the LDL-cholesterol metabolism in liver [13,79]. Another example is miR-29b found in cow’s milk, which promotes bone health by negatively regulating osteoclast differentiation and positively regulating osteoblast differentiation in humans by increasing RUNX2 activity [17]. In humans, the food-derived miRNAs, to influence gene expression, must remain unmodified until they reach their final destination. Studies of plant-derived foods have shown that miR-168 found in soybean and miR-166 found in rice remained stable after processing and cooking [74,80]. Their stability depends on several factors, one in particular is that they may be secreted and encapsulated in exosomes, which gives resistance to degradation [74]. The exosomes have special membrane characteristics that can withstand harsh conditions in the extracellular environment such as hypoxia, hypermetabolic, enzymatic and acidic environments such as that found in the stomach. This characteristic is probably due to the lipid composition of the membrane, which can change in composition based on pH modification to determine an asymmetric distribution of phosphatidylethanolamine that allows a change in lipid composition between the two membrane layers [81]. These characteristics could allow for instance vegetable miRNAs to resist the digestive phase. The exosomal miRNAs from soy, rice and milk, demonstrated their resistance to saliva, gastric, biliary and pancreatic juice digestion, thus being able to cross even the intestinal barrier [80,82]. This is impressive since the transferred miRNAs can influence target cells by actively participating in intracellular events [21]. As regards the intestinal permeability and its role in the absorption of exogenous miRNAs, there is much evidence to support the hypothesis that the microbiota plays a key role in this process and, in turn, the intestine modulates the bioavailability of nutrients [19]. Is already known that there is variability among individuals in response to physiological and pathological conditions based on age, sex, environmental factors, genetics, diet and so on. Some studies argue that there may be an interaction between miRNAs carried by extracellular microvesicles released, for instance, by plants and the gut microbiota and there are several supporting hypotheses that try to explain what happens. The first one points out that plant miRNAs might regulate intestinal permeability by regulating the microbiota gene expressions as a result of interaction with different proteins of bacteria; another one, highlights there are different bacterial mRNA targets, that facilitate their absorption from intestine [83,84]. A study performed on the plant-derived microRNA absorptions found a peak within 3 to 6 h in serum and tissues, indicating that the gastrointestinal environment is responsible for their absorption. In addition, it seems they can survive for about 36 h in the tissues and be subsequently metabolized in a similar way to endogenous miRNAs [12,85]. Exogenous miRNAs can regulate the functionality of specific bacteria in the microbiome and have an important impact on an individual’s health; it seems that, for instance, corn miRNAs can decrease the concentration of Firmicutes (an intestinal bacterium), which is correlated with an increased risk of obesity and metabolic diseases or the amount of A. Muciniphila, which increases the risk of cardiovascular disease and diabetes [20]. Interesting results emerge from studies by Chen et al., suggesting that plant-exosome-like nanoparticles from the ginger rhizome could be taken up by immune system cells (particularly macrophages) and exert immunomodulatory effects by inhibiting inflammasome activation whose activity would be associated with increased intestinal permeability and progression of diseases such as obesity [86]. Recently, a study by Cao et al. [87] also showed that ginseng-derived nanoparticles led to a reduction in melanoma tumour growth in mice by targeting macrophages and suppressing the switching of the pro-inflammatory M1 phenotype to anti-inflammatory M2 [88].

Zhao et al. suggested the reason that would hinder the detection of plant-derived miRNAs in animals would be the sequence of the miRNAs. Not all miRNAs seem to be able to be taken up, but that selective uptake would occur based on their sequence; the determining factors would be, for example, high GC content, a short length and the “CAG” motif, which could promote the uptake and delivery of these exogenous miRNAs [89].

## 8. Discussions and Conclusions

Gene activity is influenced by various external stimuli through complex mechanisms that depend on specific transcription factors and the expression of non-coding RNAs whose expression is influenced by exercise and nutrition [1,6,9,10,24,34]. In recent years, a study has also focused on understanding their influence on osteogenic processes. Factors required for osteoblast differentiation have been identified, in particular the gene expression of Runt-related transcription factor 2 (RUNX2) [14]. In particular, the miRNA whose activated pathway has been described is miR-21, which appears to play a role in promoting stem cell osteogenesis via the Smad7-Smad1/5/8-RUNX2 and Akt pathway [9,15]. Exercise as a mechanical load, plays an essential role in both the differentiation and mineralisation of osteoblasts and bone density, reducing the risk of osteoporosis [23]. Among different types of physical activity, half marathons have been shown to increase the expression of many miRNAs including miR-21-5p osteoblast differentiation and proliferation [29]. Among the micronutrients, one of the compounds that have been found to have positive effects is syringic acid (SA), a phenolic acid that on mouse mesenchymal stem cells has been shown to induce osteoblast differentiation [66]. In particular, it would appear to perform this function through increased expression of miR-21, which reduces Smad7 activity by inducing increased expression of RUNX2, with a positive effect on osteoblast differentiation [60]. It is advisable to engage in outdoor physical activity to promote vitamin D production, the intake of which is recommended along with vitamin C and silicic acid as they target the same osteogenic markers. Regarding the effect of macronutrients, Kang Gan et al. analysed the effect of high glucose levels on the regulation of osteogenic differentiation of bone marrow mesenchymal stem cells (MSCs). The results showed that increased expression of miR-221-3p and miR-222-3p on the bone tissue of diabetic mice inhibited osteogenic differentiation through activation of the IGF-1/ERK pathway [65]. Regarding the effects of protein, there are not many studies, but it would appear that a low-protein diet negatively regulates bone mass; however, there are no studies investigating the epigenetic effect of a high-protein diet. Therefore, exogenous microRNAs belonging to different species may influence human cellular processes through mechanisms that need further investigation to be better understood [12,13].

In conclusion, a correct lifestyle that includes moderate physical activity and a balanced diet, avoiding obesity conditions, are essential for the prevention of bone metabolism-related diseases because they act against abnormal epigenetic regulation. What seems clear to optimise the regenerative potential of BMSCs is to avoid low protein intake, except sulphur amino acids, and high fatty acids and carbohydrates, which negatively affect bone mass. To improve bone health, further studies looking at the modulation of bone epigenetic biomarkers due to protocols of prevention or effective alternative therapies are necessary.

## 9. Limitations of the Study

Despite the innovativeness of our study, of which no duplicates can be found in the most widely used databases, certain limitations should be highlighted. The first concerns the fact that the approach was general and did not include the study of the specific effects of different types of physical activity on miRs expression. Another limitation would be to include in the study other epigenetic mechanisms acting on bone metabolism, such as histone deacetylation and DNA methylation. In any case, the difficulties encountered are mainly because not much literature has been found to support the argument that correlates the influence of diet together with exercise on bone metabolism.

## Data Availability

Not applicable.

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
