# Peer review of "MiRNAs Expression Modulates Osteogenesis in Response to Exercise and Nutrition"

_genes, 2023, doi:10.3390/genes14091667_

Round 1
Reviewer 1 Report
This review summarises the published literature with regards to the impact of physical activity and diet on bone health. The authors describe the effects of the most potent miRs that affect bone homeostasis and their expression is influenced by nutritional diet and physical exercise.
The manuscript is well written and organised and the authors used up-to-date publications to provide details on the topic. I have only two suggestions that would improve the quality of the review: 1) since the review includes only miRs and not other epigenetic mechanisms the title should be more focused, e.g. "MiRNAs expression modulates osteogenesis...", and 2) a summative illustration that includes a summary of the text and more specifically how these miRs affect osteogenic factors would add to the work.
Author Response
We would like to thank you the referee very much for the time dedicated to the revision of our manuscript and for the constructive critique that was addressed to us.
I have only two suggestions that would improve the quality of the review:
1) since the review includes only miRs and not other epigenetic mechanisms the title should be more focused, e.g. "MiRNAs expression modulates osteogenesis...",
We agree with the referee and we changed the title as follows: “MiRNAs expression modulates osteogenesis in Response to Exercise and Nutrition”
2) a summative illustration that includes a summary of the text and more specifically how these miRs affect osteogenic factors would add to the work.
Yes, it’s true; we upload a graphical abstract to help the reader quickly focus on the topic of the review. We avoided going into too much detail by including all the miRs that were involved, to avoid creating too confusing a picture. If the reviewer considers it appropriate, we can insert the image within the manuscript.
Reviewer 2 Report
The review article by Patrizia Proia et al summarized how physical exercise, micronutrients and macronutrients affect osteogenic differentiation and bone metabolism through regulating miRNA. Overall, the writing is good and not duplicated review has been found in PUBMED. The
1. Title may be needed to be changed because it is too broad, it authors want to focus on miRNAs, the authors should change these, also miRNA is not epigenetic biomarker, they are epigenetic regulators.
2. Litteratures cited missed 2022 and 2023 publications.
3. Line 56: “Organic response” should be “organism response” or “host response”.
4. Line 90, one sentence contains two lose, please rephrase.
5. Line 119, migrates out should be released or secreted because this process is caused by cell damage, however, some miRNA are secreted by cells in the form of vesicles or exosome.
6. Line 128. “has increased the interest…..” This sentence does not have a subjective.
7. Line 130, “has identified in RUNX2 expression, “In” should be deleted.
8. Line 129-130, two “performed” in one sentence. The latter performed can be replaced with using or utilizing.
9. Line 151-240, a long paragraph, can authors split into different paragraphs for each nutrient such as vitamin D and vitamin C, OSA, Phenolic acid and others. The authors also should be certain about works of literature when they stated the different effects of micronutrients.
10. #5 Subheading should be bolded to be consistent.
11. Conclusion should not repeat some findings of literatures. It should be a summary of the findings in the field and future directions.
English is OK. Some improvement is needed.
Author Response
We would like to thank you the referee very much for the time dedicated to the revision of our manuscript and for the constructive critique that was addressed to us.
- Title may be needed to be changed because it is too broad, it authors want to focus on miRNAs, the authors should change these, also miRNA is not epigenetic biomarker, they are epigenetic regulators.
We agree with the referee and we changed the title as follow: “MiRNAs expression modulates osteogenesis in Response to Exercise and Nutrition”
- Litteratures cited missed 2022 and 2023 publications.
We took the chance to check all the references and also we did insert the reference between 2022 and 2023. In deep we added the references 25,26,40,41,49,50,63,64,75,76.
- Line 56: “Organic response” should be “organism response” or “host response”.
We have changed what you requested.
- Line 90, one sentence contains two lose, please rephrase.
We have rephrased what you requested (now line 106-108)
- Line 119, migrates out should be released or secreted because this process is caused by cell damage, however, some miRNA are secreted by cells in the form of vesicles or exosome.
We changed “migrates out” with “are release in…” now line 136
- Line 128. “has increased the interest…..” This sentence does not have a subjective.
We have changed the sentence in “Recently, there has been increased interest among researchers to identify miRNAs directly involved….” now line 145
- Line 130, “has identified in RUNX2 expression, “In” should be deleted.
Done
- Line 129-130, two “performed” in one sentence. The latter performed can be replaced with using or utilizing.
Done
- Line 151-240, a long paragraph, can authors split into different paragraphs for each nutrient such as vitamin D and vitamin C, OSA, Phenolic acid and others. The authors also should be certain about works of literature when they stated the different effects of micronutrients.
As suggested, we have divided paragraph number 5 into 4 sub-sections. In addition, we took the chance to check all the references and also did insert the references which were more relevant to the effect of the various micronutrients published between 2022 and 2023 (references 25,26,40,41,49,50,63,64,75,76).
- #5 Subheading should be bolded to be consistent.
Done
- Conclusion should not repeat some findings of literatures. It should be a summary of the findings in the field and future directions.
Thank you for the comment. Since in this paragraph we also included the future directions and also the discussions about the findings in the field, we renamed paragraph 8 as “Discussions and conclusions”
- Comments on the Quality of English Language: English is OK.
Thank you again for your suggestions.
Reviewer 3 Report
The aim of this non systematic review is such as several miRNAs are involved in the osteogenic differentiation of progenitor cells and influence some cellular processes in response to exercise and nutrition . In conclusion, a correct lifestyle that includes moderate physical activity and a balanced diet, avoiding obesity conditions, are essential for prevention of bone metabolism related diseases because they act against abnormal epigenetic regulation
The paper is interesting and novel, establishing different sections to explain the influence of nutrients and exercise on mRNA. The bibliography is adapted to the review.
Concerns
The methodology of the review should be described
One or several graphical representations of the influence of exercise and micronutrients on mRNA would help to better understand the review.
The limitations found should be included
Author Response
We would like to thank you the referee very much for the time dedicated to the revision of our manuscript and for the constructive critique that was addressed to us.
- The methodology of the review should be described
The methodology section was added as paragraph 2
- One or several graphical representations of the influence of exercise and micronutrients on mRNA would help to better understand the review.
Yes, it’s true; we upload a graphical abstract to help the reader quickly focus on the topic of the review. If the reviewer considers it appropriate, we can insert the image within the manuscript.
- The limitations found should be included
We added a “Limitation of the study” section numbered 9
Round 2
Reviewer 3 Report
The questions have answered by the authors